# Foster Kennedy Syndrome (FKS): A Case Report

**Mutali Musa** [1,2] [ID], **Gladness Aluyi-Osa** [2] **and Marco Zeppieri** [3,*] [ID]

1   Department of Optometry, University of Benin, Benin City 300238, Edo State, Nigeria; mutali.musa@uniben.edu
2   Africa Eye Laser Centre Ltd., Sapele Road, Benin City 300001, Edo State, Nigeria; gladnessaluyiosa@gmail.com
3   Department of Ophthalmology, University Hospital of Udine, 33100 Udine, Italy
*   Correspondence: markzeppieri@hotmail.com; Tel.: +39-0432-552743

**Abstract:** (1) Background: Foster Kennedy syndrome (FKS) is an ophthalmological condition characterized by an insidious reduction in vision in one eye, accompanied by clinically significant papilledema in the fellow eye. The unilateral loss of vision and optic atrophy is due to compressive optic atrophy, which causes elevated intracranial pressure that leads to swelling in the fellow eye. The risk factors for FKS include the presence of mass lesions in radiographic imaging, female gender, and increased body mass index. Differential diagnoses of FKS include tumors and pseudotumor of the frontal lobe and cranial meninges. (2) Methods: We present two cases of FKS diagnosed in February 2021 and December 2021. (3) Results: A 52-year-old male with a history of poor vision in one eye after trauma complained of constant headache. Ocular examination revealed disc pallor in his right eye with disc edema in the contralateral eye. The patient was sent for computerized tomography (CT) and placed on oral prednisolone tablets. The CT scan confirmed the diagnosis of FKS. A 30-year-old female presented to the emergency department for poor vision in her left eye and headache on the left side. Medication included dexamethasone, chloramphenicol, timolol eyedrops, furosemide, and anti-oxidant tablets dispensed from a previous private eye clinic. Ophthalmoscopy showed disc pallor with 0.1 cupping and arteriolar attenuation in both eyes with macular hemorrhages in her left eye. Bilateral papilledema secondary to raised intracranial hyper-tension was suspected. CT scans showed an intracranial mass. (4) Conclusions: These two cases show the importance of ocular examination in the diagnosis of serious systemic conditions. A concise case history, extensive ocular workup, and cranial imaging with magnetic resonance imaging and/or CT scans are indicative of patients showing acute visual loss and retro-orbital pain, which can give rise to the diagnosis of sight-threatening, permanent and fatal conditions, such as FKS. Non-surgical treatments include oral steroidal therapy, radiotherapy, and chemotherapy; however, neurosurgery is normally required.

**Keywords:** Foster Kennedy syndrome (FKS); vision loss; papilledema; optic atrophy; intracranial hypertension

## 1. Introduction

The proximity of the eye to the brain and the direct connection between the organs gives a unique ability to the examiner to observe certain ocular pathological conditions that may be located in the brain [1]. Foster Kennedy syndrome (FKS) is a rare condition in which two different findings of the optic nerve present simultaneously, mainly edema and atrophy, which could be due to a brain lesion that needs to be considered in the differential diagnosis in these cases. FKS is caused by a intracranial mass by definition, which mechanically compresses the optic nerve and causes atrophy in one eye, and induces elevated intracranial pressure that leads to papilledema in the contralateral eye.

The optic nerve transmits visual stimuli from the eye to the brain. The nerve fibers from both eyes selectively decussate at the optic chiasm. This means a disease in one eye can present with a contralateral defect. The diseases of the optic nerve are usually insidious

and have different progression rates in different eyes [2]. Loss of vision in the optic nerve diseases is common, however, does not usually present in the early stages. Vision loss often tends to be permanent. Optic nerve diseases can be unilateral or bilateral [3]. Common optic nerve disorders include coloboma of the optic nerve, neuromyelitis optica, optic nerve atrophy, ischemic optic neuropathy, optic nerve drusen, and optic neuritis.

Classical presentations of optic nerve diseases include relative afferent pupillary defects, disc pallor, disc edema, disc hemorrhages, and thinning of the neuro–retinal layer [4]. Dilated fundoscopy is indicated in all of the adults examined in the eye clinic, especially for acute symptoms and vision loss, that is, if there are no contraindications, such as closed-angle glaucoma or certain infectious diseases. A thorough eye examination in patients with acute vision loss and specific signs and symptoms, such as optic nerve atrophy and papilledema, can lead to the diagnosis of FKS, which needs a multidisciplinary approach in the management of the intracranial lesion.

We describe two cases of FKS. The study followed the tenets of the Declaration of Helsinki. The patients provided informed consent for the clinical examinations, diagnostic testing, research use of clinical records, and the data included in the study.

## 2. Materials and Methods

A 52-year-old black male presented to the clinic in February 2021 with a complaint of frontal headache and poor vision on the right eye after trauma to that eye four months before the date of visit. The patient also had occasional mucoid discharge from both eyes for one week with slight ocular pain. The patient had no other significant associated history. The patient denied any history of systemic diseases and said he wasn't on any medications.

In December 2021, a 30-year-old female visited the eye clinic, complaining of poor vision in the left eye for about six months and headache on the left side. The onset of poor vision was gradual over the 6-month period. She was on dexamethasone, chloramphenicol, and timolol eyedrops. She was also taking furosemide tablets and ocular antioxidant capsules (Ocuvite; Bausch & Lomb, USA, containing lutein, omega-3, zeaxanthin, etc.). The patient stated that her mother noticed a change in her speech and alertness. The lady had been told that "excess fats" had affected her vision at a previous eye clinic. The patient also reported a previously unknown reaction to furosemide tablets, but could not explain the symptoms.

## 3. Results

52-year-old male

During his first ophthalmologic examination, the patient's blood pressure was 119/79 mmHg with a pulse rate of 75 beats/min pulse. Visual acuity (VA) was 6/12 in the right eye and 6/6 in the left eye with a wall-mounted digital chart. The external examination and pupillary reflexes were within normal limits. The anterior segment appeared normal. Fundoscopy revealed normal retinal features with a 0.3 cup-to-disc ratio bilaterally. Intraocular pressure were normal. Refractive findings revealed Pl-0.50 × 90 in the RE with a VA of 6/5 and +0.25 − 0.25 × 90 in the LE with a VA of 6/5.

The patient was given Nepafenac 0.1% to be used once daily for two weeks and Ciprofloxacin 0.3% eye drops to be used thrice daily for three weeks. The patient was told to visit the clinic in two months.

The patient returned for a scheduled checkup two months later, his blood pressure was 124/78 mmHg with a 77 beats/min pulse. The patient's VA was right eye: 6/12−3 and left eye: 6/9 with a best-corrected VA of 6/5 in both eyes. The external and internal examinations were normal. A diagnosis of simple myopic astigmatism and presbyopia was made. The patient was discharged from the hospital with a pair of bifocal lenses and asked to return in two months for a routine follow-up.

The patient returned to the clinic two years later with a complaint of tightness and blurry vision in the right eye. The blood pressure was 122/82 mmHg and pulse rate 72 beats/min. Entry VA was light perception in the right eye and 6/12 in the left. The

pinhole did not improve vision in both eyes. Intraocular pressure was 24 mmHg and 25 mmHg on the right and left eye, respectively, using a Goldmann applanation tonometer. The penlight test gave a sluggish pupillary reaction in both of the eyes. An internal examination revealed pallor and optic atrophy with a cup to disc ratio of 0.3 in the right eye. The left eye showed optic disc edema with a 0.3 cupping.

The patient was placed on oral Prednisolone 40 mg daily (stat) tapered to 30 mg daily for one week, then tapered to 20 mg for another week, and finally, 10 mg for the fourth week. The patient was sent for a magnetic resonance imaging (MRI) scan of the brain and for a neurology consult.

At the next follow-up two weeks later, the patient's vision had deteriorated to no light perception (NLP) in the right eye and count finger@1 m in the left eye. The pupillary exam showed a relative afferent pupillary defect of the right eye and very sluggish response in the left eye. Ocular motility was normal for both eyes.

MRI scans, which were performed eight days after the first ophthalmologic examination on 13 February 2021, showed a uniformly enhancing base of skull mass lesion, consistent with a frontal-sphenoidal meningioma with a dimension of 4.6 cm × 3.5 cm × 4.9 cm (AP × L × TS). Figure 1 shows the axial and sagittal sections of the intracranial mass that compresses the right frontal lobe and displaces the falx.

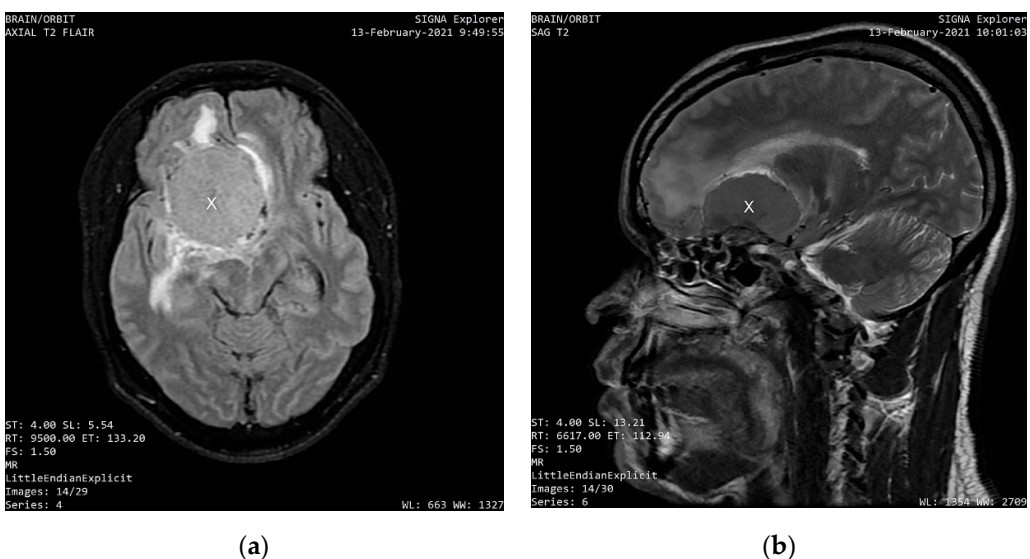

(**a**)        (**b**)

**Figure 1.** (**a**) An axial section T2 weighted scan showing a well-circumscribed, homogenous mass compressing the right frontal lobe and displacing the falx and the left frontal lobe; (**b**) A sagittal section of the head of the same patient showing the compressive mass.

The patient did not return for subsequent ophthalmologic follow-up visits. The patient underwent neurosurgery, but later passed away due to complications associated with the surgery.

30-year-old female

Entry VA was 6/12 in the right eye and hand motion in the left eye. The pupillary responses were sluggish. Intraocular pressure was 13 mmHg in the right eye and 20 mmHg in the left eye, with no improvement in VA with pin-hole. Fundoscopy revealed a bilateral 0.4 cup-to-disc ratio with pallor. She was using timolol eyedrops in both eyes for ocular hypertension (intraocular pressure before therapy was 24 mmHg in both eyes, and previous visual fields and nerve fiber analysis were normal). There was arteriolar attenuation in both eyes, with macula hemorrhages in the left eye. A C-shaped halo was visible in the left eye. An initial assessment of unilateral papilledema with contralateral optic nerve atrophy was made due to intracranial hypertension. The patient was referred for a same-day computerized tomography (CT) scan, which showed an intracranial mass (Figure 2).

This patient was immediately sent for a specialist consult at the neurosurgery department of the federal state teaching hospital. She was scheduled for emergency brain surgery immediately after the consult. The lady later passed away due to complications of the lesion and surgery.

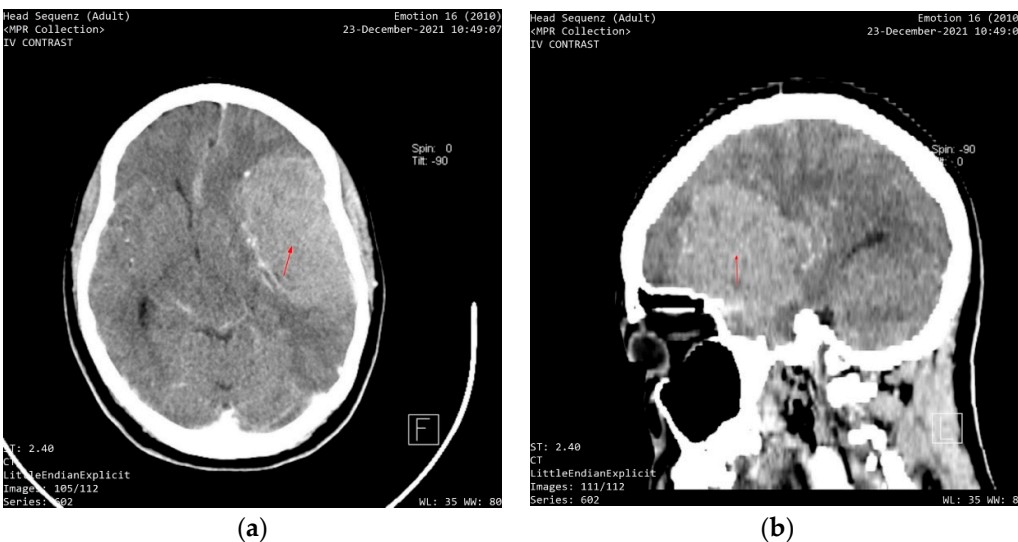

**Figure 2.** (**a**) An axial section showing a contrast image depicting a well-circumscribed, homogenous mass compressing the left frontal lobe; (**b**) A sagittal section shows a contrast image depicting an intracranial mass.

## 4. Discussion

The ocular disease conditions that have a similar presentation as those seen in patients with FKS include pseudo-foster Kennedy syndrome, non-arteritic anterior ischemic optic neuropathy (NAION), and optic nerve hypoplasia, with the classical signs of either unilateral optic atrophy or unilateral papilledema. Non-arteritic anterior ischemic optic neuropathy is the second highest cause of optic neuropathy [5]. It is a non-inflammatory ischemia that typically occurs about 1 mm from the optic disc, caused by a reduction in blood flow through the posterior ciliary artery which supplies the optic nerve head (or the anterior part of the optic nerve). It was suggested to be caused by an embolism. NAION can also be caused by a nocturnal drop in perfusion pressure.

Optic nerve hypoplasia is a congenital condition in which the optic nerve does not develop fully, thus giving rise to the typical small and pale appearance seen in optic nerve hypoplasia. There are cases in which the optic nerve hypoplasia is seen together with NIAON [6]. The causes are not well known, however, there have been correlational studies showing the relationship between optic nerve hypoplasia and maternal intake of quinine, phenytoin, lysergic acid diethylamide, and fetal alcohol syndrome [7,8].

Optic neuritis is the inflammation of the optic nerve [9], and is called optic papillitis when it involves the optic nerve head; neuro-retinitis when it involves the optic nerve head and the macula; while it is called retrobulbar neuritis when it involves the posterior part of the nerve.

Papilledema is a sign of a pathologic condition in the eye, presenting as an elevation of the optic nerve head. It results from a sustained increase in the intracranial pressure, in which the eye presents with varying degrees of blurred optic disc margin and reduction.

FKS typically presents with optic atrophy in one eye and contralateral papilledema (Type 1). Other types of FKS have been discussed, which include Type 2 in which there is bilateral papilledema with unilateral optic atrophy, and Type 3 that shows bilateral papilledema developing into bilateral optic atrophy. A condition that presents similarly to the type 1 FKS is pseudo-Foster Kennedy syndrome, in which the only difference is usually the cause, which is not by an intracranial lesion but by anterior ischemic optic neuropathy.

The compressing mass/tumor is often an anterior cranial fossa meningioma (this may be in the olfactory groove, sphenoid wing, or frontal lobe), although the middle cranial fossa can also be involved. When the mass is large enough, it can cause an increase in the intracranial pressure, which leads to contralateral papilledema. As the disease progresses, the ipsilateral atrophic optic nerve loses nerve fibers and thus does show less signs of papilledema.

FKS usually presents with large tumors of the frontal lobe and olfactory groove meningiomas. Smaller tumors, while equally morbid, may not present with the same compressive effects on the orbit when compared to the larger tumors. The meninges are the protective tissues that cover the brain and the spinal cord. Meningiomas are space-occupying lesions with an origin in the central nervous system that develop when the meninges begin to grow abnormally. Tumors of the sphenoid wing and pituitary gland may also cause the condition. Pseudo-FKS, however, typically presents with the classical ocular hallmarks of FKS in the absence of a causative space-occupying tumor. Conditions such as diabetic papillopathy, unilateral hypoplasia of the optic nerve, non-arteritic anterior ischemic optic neuropathy (NAION), and compression of the optic nerve by the gyrus rectus muscle can cause pseudo-FKS.

The risk factors for FKS include neurofibromatosis type 2, female gender (the role of hormones has been suggested as a possible explanation for it occurring more frequently in females), high body mass index, and prior exposure to cranial radiation. With regards to diagnosis, FKS requires a detailed case history. The patient should be questioned on vision and sudden reduction or gradual loss of vision. The history should involve probing to elicit the signs of increased intra-cranial pressure, which may include headache, diplopia, nausea, and vomiting. Other symptoms usually noticed by patients include anosmia, diplopia, and emotional disorientation. The family members of patients should also be questioned with regards to any changes in the behavior of the patient.

The signs usually detected by the clinician include ipsilateral optic atrophy and contralateral papilledema as seen on fundus examination, a relative afferent pupillary defect in the eye with the optic atrophy, reduced VA, especially in the eye ipsilateral to the location of the lesion diplopia, due to increased intracranial pressure and visual field loss. In most cases, the VA of the contralateral eye is usually spared until the advanced/late stages of the condition. There may also be an evident proptosis, if there is orbital involvement of the tumor, especially if the lesion is located in the anterior cranial fossa. Clinical diagnosis is typically by neuroimaging studies; a head and orbital CT scan and/or MRI with or without contrast should be completed in all of the patients with suspected FKS.

The management of FKS is usually by surgical resection, although chemotherapy and radiotherapy [10] are also considered in elderly geriatric patients who are more prone to mortality with invasive surgery. With regards to medical therapy, the first line of action is typically with corticosteroids [11], which is usually an asymptomatic treatment for the reduction of edema around the tumor and intracranial pressure. In older patients, it is preferable to go for a non-neurosurgical procedure, such as stereotactic radiotherapy [12] and radiosurgery. The use of hydroxyurea has been anecdotally found to be useful, especially for unresectable tumors and large residual tumors [13].

## 5. Conclusions

FKS is a rapidly evolving, insidious condition. Routine examinations, such as dilated fundoscopy and ophthalmoscopy, in addition to a proper case history can make a great difference in the detection and prognosis of the condition. An MRI is the best diagnostic option for suspected cases. Ocular signs and symptoms may be accompanied by systemic presentations, such as anosmia, nausea, and emotional imbalance. This fatal condition results in ocular morbidity and can progress to total blindness in both eyes. In extreme cases, the condition is life-threatening. The management of FKS requires a coordinated multi-disciplinary approach. The prognosis of FKS depends largely on the extent of the intracranial space-occupying lesion.

**Author Contributions:** Conceptualization, M.M. and G.A.-O.; methodology, M.M. and G.A.-O.; validation, M.M. and M.Z.; formal analysis, M.M. and M.Z.; investigation, M.M.; resources, M.Z.; writing—original draft preparation, M.M.; writing—review and editing, M.M. and M.Z.; visualization, M.M. and M.Z.; supervision, M.M.; project administration, M.M. All authors have read and agreed to the published version of the manuscript.

**Funding:** This research received no external funding.

**Institutional Review Board Statement:** Ethical review and approval for this study was waived due to the observational nature of the study. Patients provided consent in medical records to the disclosure of data in an anonymous form for research and publication purposes.

**Informed Consent Statement:** Informed consent was obtained by the patients.

**Acknowledgments:** The authors thank Godwin Okoye, Founder of Africa Eye Laser Centre Ltd., Nigeria.

**Conflicts of Interest:** The authors declare no conflict of interest.

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
