# Peer review of "Foster Kennedy Syndrome (FKS): A Case Report"

_clinpract, doi:10.3390/clinpract12040056_

Round 1
Reviewer 1 Report
This is a presentation of two cases of alleged Foster-Kennedy syndrome.
This cannot be a case series, since there are only two reported patients, and for case series there should be four or more patients.
The authors put effort into this manuscript, but I personally have some dilemmas:
PAGE 1 LINE 39 „different diseases of the optic nerve present simultaneously “ I would rather say two different findings or presentations on optic disc, since one of them is edema, and the other is atrophy.
PAGE 2 LINE 73 “ocular antioxidant capsules” – please state precisely the name of those capsules.
PAGE 3 LINE 89 Why blood pressure data is relevant in this case? We do not have the data on this the first time he presented, only for follow-ups.
PAGE 3 LINE 92 “presbyo-pia” change to presbyopia
PAGE 3 LINE 93„. The patient was dispensed with …” change to discharged from the hospital.
When was MRIdone, in relation to the time of admission?
PAGE 3 LINE 132 “An initial assessment of bilateral papilledema secondary to raised intracranial hypertension” – if she had bilateral papilledema, than it is not Foster-Kennedy syndrome. Change “raised intracranial hypertension” if it is raised, than it is hypertension…
PAGE 3 LINE 132 „bilat-eral” change to bilateral
Why was the second patient on antiglaucoma medication? At her age (30 years), and with CD ratio of 0.1?
Reviewer 2 Report
- Line 56: should be "infectious"
- Line 77: perhaps: "could not explain the symptoms"
- A discussion about the typical tumors associated with FKS would be good.
- A further discussion about pseudo-FKS with subsequent NAION would be good.
Round 2
Reviewer 1 Report
Thank you for accepting my suggestions. Best regards